# Classical and Next-Generation Vaccine Platforms to SARS-CoV-2: Biotechnological Strategies and Genomic Variants

**DOI:** 10.3390/ijerph19042392

**Published:** 2022-02-18

**Authors:** Rachel Siqueira de Queiroz Simões, David Rodríguez-Lázaro

**Affiliations:** 1Institute of Technology in Immunobiologicals, Bio-Manguinhos, Oswaldo Cruz Foundation, Fiocruz, Avenida Brasil, 4365, Manguinhos, Rio de Janeiro 21040-900, Brazil; rachel.queiroz@bio.fiocruz.br; 2Microbiology Division, Faculty of Sciences, University of Burgos, 09001 Burgos, Spain; 3Research Centre for Emerging Pathogens and Global Health, University of Burgos, 09001 Burgos, Spain

**Keywords:** genomic variants, technological platforms, SARS-CoV-2

## Abstract

Several coronaviruses (CoVs) have been identified as human pathogens, including the α-CoVs strains HCoV-229E and HCoV-NL63 and the β-CoVs strains HCoV-HKU1 and HCoV-OC43. SARS-CoV, MERS-CoV, and SARS-CoV-2 are also classified as β-coronavirus. New SARS-CoV-2 spike genomic variants are responsible for human-to-human and interspecies transmissibility, consequences of adaptations of strains from animals to humans. The receptor-binding domain (RBD) of SARS-CoV-2 binds to receptor ACE2 in humans and animal species with high affinity, suggesting there have been adaptive genomic variants. New genomic variants including the incorporation, replacement, or deletion of the amino acids at a variety of positions in the S protein have been documented and are associated with the emergence of new strains adapted to different hosts. Interactions between mutated residues and RBD have been demonstrated by structural modelling of variants including D614G, B.1.1.7, B1.351, P.1, P2; other genomic variants allow escape from antibodies generated by vaccines. Epidemiological and molecular tools are being used for real-time tracking of pathogen evolution and particularly new SARS-CoV-2 variants. COVID-19 vaccines obtained from classical and next-generation vaccine production platforms have entered clinicals trials. Biotechnology strategies of the first generation (attenuated and inactivated virus–CoronaVac, CoVaxin; BBIBP-CorV), second generation (replicating-incompetent vector vaccines–ChAdOx-1; Ad5-nCoV; Sputnik V; JNJ-78436735 vaccine-replicating-competent vector, protein subunits, virus-like particles–NVX-CoV2373 vaccine), and third generation (nucleic-acid vaccines–INO-4800 (DNA); mRNA-1273 and BNT 162b (RNA vaccines) have been used. Additionally, dendritic cells (LV-SMENP-DC) and artificial antigen-presenting (aAPC) cells modified with lentiviral vector have also been developed to inhibit viral activity. Recombinant vaccines against COVID-19 are continuously being applied, and new clinical trials have been tested by interchangeability studies of viral vaccines developed by classical and next-generation platforms.

## 1. Introduction

Coronaviruses (CoV) are divided into four genera: Alpha (α), Beta (β), Gamma (γ), and Delta (δ-CoV) (Figure 1). Seven CoVs have been identified as being able to infect humans. The α-CoVs HCoV-229E and HCoV-NL63 and the β-CoVs HCoV-HKU1 and HCoV-OC43 cause mild respiratory infection. Severe acute respiratory syndrome coronavirus (SARS)-CoV-2 can cause severe disease [1] and is a β-coronavirus, subgenus *Sarbecovirus*, *Orthocoronavirinae* subfamily, *Coronaviridae* family [2,3].

Bats are reservoirs for a wide variety of coronaviruses, including SARS-CoV and Middle Eastern respiratory syndrome coronavirus (MERS-CoV) viruses. SARS-CoV emerged in 2003 from recombination between bat (genus *Phinolopus*) coronaviruses and started to circulate in intermediate hosts, notably civets (*Paguna larvata*), a common carnivore in Asia. These viruses continued to cross species barriers to adapt to humans. In 2012, MERS-CoV from bats adapted to dromedaries and started to infect humans [2,4,5].

Ecological and environmental factors facilitate the emergence of new pathogens, such as SARS-CoV-2, and their spread to several animals’ species able to be reservoirs (natural and intermediate hosts) of infectious diseases. The interactions between humans, animals, and environment across large-scale geographic barriers induce zoonotic spillover events causing ecological and socioeconomic impacts in the one health approach. Biological invasions by emergent zoonotic viruses across geographic barriers through interfaces between reservoir hosts, intermediate hosts, and humans have been described in susceptible populations [6,7]. The origin of SARS-CoV-2 involved zoonotic transfer, independent of whether the source of transmission was an animal host such as Malayan pangolins (*Manis javanica*) or previous natural selection in humans or laboratory escapes of SARS-CoV during cell culture passage or from animal models [1].

SARS-CoV, MERS-CoV, and SARS-CoV2 genome sequences show that they are phylogenetically close. The current outbreak of acute respiratory disease associated with SARS-CoV-2 started in Wuhan, China, and spread rapidly throughout the world. However, the virus has been detected in sewage samples in southern Brazil (November 2019), demonstrating that the virus was circulating more than two month before public health notifications [8].

Animal experiments as biological model systems and study design strategies for classical and next-generation vaccines are key points of safety and efficacy in randomized and non-randomized clinical trials in different technological platforms using first-generation (attenuated and inactivated virus-CoronaVac, CoVaxin; BBIBP-CorV), second generation (replicating-incompetent vector vaccine -ChAdOx-1; Ad5-nCoV; Sputnik V; JNJ-78436735 vaccine, replicating-competent vector, protein subunits, virus-like particles-NVX-CoV2373 vaccine), and third-generation vaccines (nucleic-acid vaccines–INO-4800 (DNA); mRNA-1273 and BNT 162b (RNA vaccines).

## 2. Cellular and Humoral Immunity

Transcriptome analysis has confirmed that NK cell counts are decreased in the peripheral blood of patients with severe COVID-19. CXCR3 and CXCL9-11 ligands are increased by SARS-CoV-2 in lung tissue in vitro, and SARS-CoV-2 mediates the recruitment of peripheral blood NK cell infiltration into the lungs in infected patients [9]. Immunological memory is still an area with many questions to be elucidated. Until now, previous studies suggested that T cells may confer long-term immunity after, identifying that virus-specific T cells persisted for at least 6–11 years. Virus-specific CD4 and CD8 T cells detected in infected patients were characterized by CD45RA and CCR7 expression as CD4 Tcm (central memory) or CD8 Tem (effector memory). Virus-specific T cell responses as the production of specific inflammatory cytokines by CD4 T cells correlated with Th2 cell (IL-4, IL-5, IL-10) serum cytokines are increased in cases of severe disease [9]. The humoral immune response, unlike cellular immunity, can be transmitted by plasma or serum such as the production of hyperimmune globulins in immunize horses with recombinant trimeric spike (S) glycoprotein against SARS-CoV-2. Plasma from immunized animals showed titers 150-fold higher than the neutralizing titers in human convalescent plasma as a new therapeutical against COVID-19 [10]. The weakened immune system promotes the proliferation of opportunistic invading pathogens such as the virus Epstein–Barr virus (EBV), a human gamma herpesvirus. EBV reactivation induced by COVID-19 inflammation may occur soon after or concurrently with long COVID-19 infection [11].

Another study about diabetic individuals with severe lung lesions analyzed monocyte metabolism that modulates T cell response to SARS-CoV-2 and identified several glycolysis-associated genes upregulated in monocytes [12]. The virus replicates in vitro more at high concentrations of glucose in monocytes infected. Consequently, high glucose levels in diabetic patients captured by monocytes favor massive virus replication and the release of large amounts of cytokines. The cytokine storm, including the interleukins IL-6 and IL-1 β, and tumor necrosis factor α, generates an uncontrolled inflammatory response leading to systemic changes directly affecting lung cells [12].

Patients severely ill with COVID-19 have been found to display overexpression of genes involved in the INF-α and INF-β signaling pathway, which is linked to the antiviral response [12]. In Vero cell culture, both IFN-a and IFN-β reduce the viral load, and IFN-β showed greater potential for action against the virus. IFN-α decreased the viral titer by 3.4 log of IFN-β by above 4 log, and treatment with both cytokines prevented cytopathic effects on Vero cells [13].

## 3. Phylogeny and Viral Genomics

SARS-CoV-2 is an enveloped positive-sense, single-stranded RNA (ssRNA) coronavirus, presenting as spherical particles of 100–160 nm in diameter with a genome of 27–32 kb. The 5′-terminal two-thirds of the RNA genome is polycistronic and encodes a polyprotein, pp1ab, mainly in open reading frame 1 (ORF-1). This protein is cleaved into 16 non-structural proteins (NSPs1–NSPs16). The 3′ terminus one-third of the SARS-CoV-2 genome is partly monocistronic and encodes four essential structural proteins, called spike (S), envelope (E), membrane (M), and nucleocapsid (N) [2]. S-glycoprotein of SARS-CoV-2 can bind to host cell receptors for virus entry by angiotensin-converting enzyme 2 (ACE2) on the surface of human cells. S glycoprotein includes two subunits, S1 and S2. S1 determines the virus–host range and cellular tropism by RBD domain, while S2 mediates virus–cell membrane fusion by two tandem domains, heptad repeats 1 (HR1) and heptad repeats 2 (HR2). M protein is responsible for the transmembrane transport of nutrients, the bud release, and the formation of viral envelope. Both nucleocapsid and envelope protein interfere with host immune response [2]. The three-dimensional structure of the S protein has been determined by computer modelling and cryoelectronic microscopy techniques [14]. Molecular dynamic simulations and computer docking analyses have been used to develop drugs based on the SARS-CoV-2 structure. Physical equations are used to predict interactions between therapeutic targets and potential drugs to identify agents that can subsequently be tested for inhibition of viral infectivity in cell culture and animal models [15].

## 4. Genomic Variants

New genomic variants in the SARS-CoV-2 spike facilitate interspecies transmissibility and adaptation of animal strains to humans. Recombination involving whole genes or genomic fragments and the exploitation of cellular genes in the viral replication process make infection of new species possible. This phenomenon of a virus adapting to a new host is described as spillover [5,6]. The continued evolution of the virus in an animal reservoir could potentially lead to recurrent animal SARS-CoV-2 to humans spillover events [16].

A phylogenetic tree of isolates from bats, pangolins, and humans, based on the receptor-binding motif (RBM) of the spike protein and other genes has been established. SARS-CoV-2 shares a high level of genetic similarity (96.3%) with CoV Ra T013, isolated from a bat in Yunhan in 2013 [17]. Pangolin SARS-like CoVs fall into two distinct groups, corresponding to the places of origin: an isolate in the first branch, Pan_Sl-CoV_GD, from the Guangdong (GD) province of China that shows high genetic similarity to SARS-CoV-2 (91.2%) and a sample from the second branch, Pan_SL-CoV-GX, isolated in the Guagxi province (GX) shows 85.4% [17]. SARS-CoV and SARS-CoV-2 viruses, despite using the same ACE 2 receptor, showed significant genetic divergence. However, the structures of the S1 unit of the S protein in the two viruses are highly similar except for the loop that folds differently [17] (Figure 2).

Since the beginning of the pandemic, 4000 genomic variants have been observed. However, 80% of the genomic variants are synonymous (no change in the amino acid encoded). Non-synonymous genomic variants where amino acid substitution or deletion occurs, are more significant. The first genomic variants in the SARS-CoV-2 spike glycoprotein were registered in January 2020 in China: D614G (aspartate (D) at amino acid position 614 replaced by glycine (G). About 20% of Brazilian isolates in May 2020 had this D614G. All Brazilian isolates in February 2021 presented the D614G mutation [16].

In October 2020, a new variant (N501Y) called B.1.1.7 was detected in the UK [1]. In December of the same year, a new variant called B.1.351 was detected in South Africa (501Y.V2) [16]. Both variants (501Y.V2 and 501Y) carried another mutation: E484K. K417N/E484K/N501Y variants have since been isolated in Japan, the USA, France, and Italy. In December 2020, a new variant was found in Manaus, in Brazil, called (P1) B.1.1.28. In the following month, in January 2021 a new variant called P2 (E484K) was detected in Rio de Janeiro, but it is less aggressive than P1 [1].

One of the most variable parts of the SARS-CoV-2 genome is the encoding of the receptor-binding domain (RBD) in the spike protein. The RBD of SARS-CoV-2 binds specifically to the ACE2 receptor in humans and other host animal species [1]. Six RBD amino acids have been shown to play a key role in binding to ACE2-Y442, L472, N479, D480, T487, and Y4911, which corresponds to L455, F486, Q493, S494, N501, and Y505 in SARS-CoV-2 [1]. During viral replication, there can be changes in the genomic nucleotide sequence, and such errors are more common viruses with RNA than DNA genomes. These replication errors are a major source of genetic variability, and the genetic diversity of SARS-CoV-2 and in the gene encoding the spike protein may be in part the consequences (Figure 2).

PROVEAN software (Protein Variation Effect Analysis) has been used to analyze all known genomic variants (synonymous and non-synonymous) in SARS-CoV-2 proteins, and 536 mutated positions have been identified among the variants from Indian isolates where the proteins ORF1ab, spike (S), nucleocapsid (N), ORF3a, ORF7a, and ORF8 are most susceptible of genomic variants [18]. A major concern is the emergence of new variants resistant to available vaccines, particularly if associated with greater transmissibility and virulence.

Viral neutralization assays and analyses of plasma from convalescent individuals have demonstrated the persistence and resistance of new variants of SARS-CoV-2. Nevertheless, some proteins that could serve as targets of vaccines are conserved in SARS-CoV-2 variants. Additionally, the nucleocapsid (N) protein is conserved in all coronaviruses (SARS-CoV-2, SARS-CoV, MERS-CoV). Cryo-EM structural investigations of the SARS-CoV-2 N protein (48 kDa) have revealed their interactions with human antibodies. However, the N proteins in other coronaviruses such as OC43, HKU1, NL63, and 229E show structural divergences [19].

Coronavirus variants can be classified into three categories: variants of interest (VOI), variants of concern (VOC), and variants of high sequence (VOHC) [20]. Despite the high rates of transmissibility and characteristics of genomic variants, until this moment, no SARS-CoV-2 variants have been designated as VOHC [20]. The strains from the United Kingdom (B.1.1.7), currently classified by World Health Organization (WHO) label as Alpha; from Brazil at Manaus city (P.1), classified as Gamma; from South Africa (B.1.351 or 501.V2), classified as Beta; and B.1.617.2 first identified from India and classified as Delta by WHO are variants of concern. The variants’ VOI and VOC are shown in Table 1 and Table 2, respectively.

### 4.1. B.1.1.7

Lineage B.1.1.7 carries at least 17 amino acid genomic variants with changes at 23 nucleotide positions in the genome. Eight genomic variants map to the spike glycoprotein, including N501Y in the receptor-binding domain, deletion 69_70, and P681H in the furin cleavage site [16,21]. The strain is 70% more transmissible than other SARS-CoV-2 strains [21]. This increased infectivity is also associated with the molecular interactions between the mutant residue Y501 in the spike RDB and the residues Y41 and K353 of the host ACE2 receptor [22].

### 4.2. B.1.351

This variant was found in South Africa in October 2020 and is also called the 501Y.V2 [21]. B.1.1.351 has three genomic variants (K417N, E484K, and N501Y) in the RBD and is more transmissible than the parental strain; it shows an increased viral load and resistance to neutralization by antibodies generated either by natural infection or by vaccination. Structural modelling has been used to study interactions of mutation site residues and the antibody receptor-binding domain (RBD) [23]. B.1.351 is neutralized by convalescent plasma and serum from individuals vaccinated with the Pfizer–BioNTech BNT162b2 vaccine with an effectiveness similar to that of the parental strain. However, the Janssen single-dose COVID-19 vaccine and Oxford–AstraZeneca AZD1222 vaccine were less effective against this variant than against the control strain [23].

### 4.3. P1 and P2

The P.1 variant, also known as B.1.1.28, was identified in November 2020 in Manaus. It carries 17 unique amino acid changes, three deletions, and four synonymous genomic variants [24]. The E484K, K417T/N, and N501Y genomic variants are the most worrisome because they generate greater transmissibility by increasing viral contagion [25,26]. The P1 variant is highly resistant to previously developed antibodies, and it can reinfect individuals who have already had COVID-19. The P.1 variant was detected in a screening nasopharyngeal swab sample from an asymptomatic traveler from Brazil (GenBank accession no. MW517286) [26]. The P.2 lineage reported in Rio de Janeiro, Brazil, only carries the E484K genomic variants [25].

### 4.4. B.1.617

Variant B.1.617, detected in India, has 13 genomic variants, including a “double mutant” in the spike protein (E484Q and L452R). It carries the common genomic variants D111D, G142D, D614G, and P681R in the spike protein, including the receptor-binding domain (RBD). There are three versions of this variant (B.1.617.1, B.1.617.2, and B.1.617.3), and it has been found in more than 51 countries. It received the classification of variant of interest, and it is now considered by the World Health Organization as a “variant of concern” due to its higher transmissibility. The enhanced transmissibility is in part due to increased ACE2 binding and rate of S1–S2 cleavage by RBD associated with genomic variants L452R with P681R in the furin cleavage site [26]. The most recent reported occurrences in different countries on the European continent (Table 3), Asian continent (Table 4), American continent (Table 5), Oceanian continent (Table 6), and African continent (Table 7), according tracking of variants (VOC Delta GK (B.1.617.2 + AY) first detected in India, and the most recent reported occurrences in different countries (VUM GH/490R (B.1.640 + B.1.640*) first detected in Congo/France are shown in Table 8 below (adapted https://www.gisaid.org/hcov19-variants/, accessed 5 January 2022).

### 4.5. B.1.1.529

A new variant of SARS-CoV-2 was first detected in specimens collected on 11 November 2021 in Botswana and on 14 November 2021 in South Africa. The World Health Organization (WHO) classified B.1.1.529 Omicron as a variant of concern (VOC) on 26 November 2021. The Omicron variant is characterized by at least 30 amino acid substitutions, of which 15 of them are in the receptor-binding domain (RBD). Moreover, three small deletions and one small insertion occur in other genomic regions [20]. Due to the deletion at H69 and V70 in the B.1.1.529 (Omicron) spike protein, TaqPath COVID-19 Combo Kit (Thermo Fisher) designed to detect multiple genetic targets has been used in detection of new genomic variants and should be confirmed by sequencing [20]. The map of phylodynamics of pandemic coronavirus variant VOC Omicron GRA (B.1.1.529+BA) showing 3.985 of 3.985 genomes was last updated 19 December 2021 by tracking of variants on GISAID (Figure 3). Relative variant genome frequency per region and map of tracked variant occurrence can be reported in real time on https://www.gisaid.org/hcov19-variants/, accessed on 5 January 2022. The new variant has already been detected in 94 countries that shared 146,843 Omicron genome sequences, and new data are constantly being updated by the genomic surveillance platforms of the SARS-CoV-2 virus. Currently, Omicron is classified into four lineages (BA.1, BA.1.1, BA.2, and BA.3), with BA.1 being the most prevalent in the number of infected cases.

### 4.6. B.1.640.2

A new variant has been detected in southeastern France by qPCR assays and consists in 46 nucleotide substitutions and 37 deletions, resulting in 30 amino acid substitutions and 12 deletions (9/12 are in the spike protein). About 14/30 substitutions including N501Y and E484K are presented in other variants as the Beta, Gamma, Theta, and Omicron. The B.1.640 lineage corresponds to a variant first detected in France (April 2021), in Indonesia (August 2021), and in Republic of the Congo (September 2021). Some changes between both lineages are that the spike genes differ by seven mutations, and more 25 nucleotide substitutions and 33 nucleotide deletions have been detected. It is necessary to assess whether commercially available vaccines are able neutralize a new variant or whether the new genomic variant induces viral escape [27].

### 4.7. Other Variants

VOC-202102/02 (Bristol) and VUI-202201/01 (Liverpool), both found in the UK, carry 484K genomic variants and can escape antibodies induced by vaccines. Another variant, B.1.525, also has the 484K. The P1 and P2 variants are derived from the B.1.1.28 lineage, and the N9 variant originates from the B.1.1.33 lineage. These three variants contain the E484K present in all regions of Brazil. Variant B.1.429 from California, United States, shows genomic variants in L452R and can escape antibodies. Other variants of interest include B.1.525 that emerged in UK, B.1.1.49 that emerged in Denmark, and A.23.1 that was first detected in Uganda [26].

## 5. Diagnostics

Biopharmaceutical and biotechnology companies have developed oligonucleotides and tests for detection of SARS-CoV-2 directed at three target genes coding for the envelope (E), nucleocapsid (N), and RNA-dependent RNA polymerase (RdRP) protein. Quantitative reverse transcriptase polymerase chain reaction (qRT-PCR) and new-generation sequencing platforms have been used as standard tools in viral diagnosis. qRT-PCR has detected continued virus infection in patients 42 days post-infection. Transcriptome analyses involving RNA sequencing has shown that the counts of natural killer cells, which recognize the virus through cell receptors, are lower in the peripheral blood of patients with severe COVID-19 than that of controls.

Viral neutralization assays with pseudo virus production have been tested using African green monkey kidney Vero cells, and SARS-CoV-2 viral stocks have been propagated in the same type of cell culture [12]. New tests for antibody binding to cell surface spike proteins of SARS-CoV, SARS-CoV-2, and MERS-CoV have been detected by immunofluorescence microscopy and low cytometry assay based on the HEK-293 T cell line transfected with viral plasmids fused to green fluorescence protein (GFP) using lipofectamine [10]. Viral neutralization assays for SARS-CoV-2 by plaque reduction test (PRNT) techniques have been used to determine viral titers, and enzyme-linked immunosorbent assay and immunofluorescence microscopy have been used to study antibodies.

The three proteins that are targets of the main diagnostic tests for COVID-19 are the envelope (E), nucleocapsid (N), and RNA-dependent RNA polymerase (RdRP) proteins. Several methods for detecting IgG and IgM antibodies and antigens of the SARS-CoV-2 virus have been developed for point-of-care tests, and they generally involve fluorescence and immunochromatography assays.

Many chimeric monoclonal anti-SARS-CoV-2 antibodies have been developed based on the S protein RBD epitope, which was developed by ACRObiosystems (antibody S1N-M122) and was able to neutralize the variants identified in the United Kingdom (B.1.1.7), South Africa (B.1.351), Brazil (P.1), and California (B.1.429), and the wild-type variant carrying D614G (WT-D614G). This S1N-M122 antibody will be valuable for quality control for recombinant vaccines, as a positive control in screening for neutralizing antibodies, and for COVID-19-antigen testing assays.

## 6. Viral Biosynthesis

The high transmissibility and virulence SARS-CoV-2 are in part explained by the interactions between the spike protein and transmembrane serine protease 2 enzymes (TMPRSS2), cathepsins, and furin present in large quantities in human cells. These three enzymes act as genetic scissors, cutting and thereby activating the spike protein; this facilitates the fusion of the viral membrane into the cell membrane [28].

Coronaviruses are coated with a group of proteins known as spike proteins or glycoprotein S (150 KDa). Host transmembrane serine protease enzyme type II (TMPRSS2) activates S proteins homotrimers by interaction with residues 331 to 524 [29]. After activation, the S protein binds to the host cell angiotensin-converting enzyme 2 (ACE-2), which serves as the receptor allowing cell entry [2] and is then cleaved by the furin-like host cell protease into two subunits, S1 and S2. ACE-2 regulates vasoconstriction and vasodilation through the action of renin, angiotensin, and androsterone. ACE-2 is highly expressed in cells of the lower respiratory tract, cells of the gastrointestinal tract, stratified epithelial cells, enterocytes, cardiomyocytes, hepatocytes, and urinary cells [30].

There has been work to identify other host factors involved in cell entry. CD 47 may anchor the virus, promoting fusion of the virus and the cell membrane, release of the viral RNA, or formation of a complex containing the virus; this may favor endocytosis and consequently the release of viral RNA into the cytoplasm [30].

Most of the synthesized nonstructural proteins (nsps) form a replicase–transcriptase complex in double-membrane vesicles in the host cell, which are mainly an assembly by RNA-dependent RNA polymerase (RdRp) and helicase. Moreover negative-sense genes and positive-sense mRNAs are transcripted by a complex using the genome template of viral entry [29]. Subgenomic fragments are translated into structural and accessory proteins including M, S, and E, which are then anchored to the endoplasmic reticulum and progress to the endoplasmic reticulum–Golgi intermediate compartment. Finally, replicated RNA is incorporated into this viral envelope, and new viruses sprout through the Golgi complex, where the nucleocapsids bind other structural proteins and form small vesicles; the vesicles export the new infectious viral particles by exocytosis [30].

## 7. Classical and Next-Generation Vaccine Platforms

Diverse approaches are being used for the development of vaccines against COVID-19: live attenuated virus, inactivated vaccines grown in cell culture, non-replicating viral vectors such as adenovirus in particular type 5 adenovirus (Ad5-nCoV) (CanSino Biologicals Company), expressing protein S carrying SARS-CoV-2 particles [31] or replication-incompetent vector vaccines, replicating or replicating-competent viral vector vaccines expressing the S protein, recombinant protein (recombinant spike protein-based and recombinant RBD-based vaccines), peptide-based, virus-like particle (VLP), nucleic-acid vaccines (use the electroporation of plasmid DNA that encodes the virus S protein and mRNA vaccines encapsulated in Lipid nanoparticles (LNP) that encodes the S protein for example Modern Company with mRNA technology-1273), and dendritic cells modified and artificial antigen-presenting cells, both modified with lentiviral vector [31,32,33,34]. Technological platforms for SARS-CoV-2 vaccine candidates licensed or under development are shown in Table 9.

### 7.1. Weakened and Inactivated Virus

Vaccines using live-attenuated virus based on by a weakened version of the flu virus adapted to COVID-19 virus have been developed by the University of Hong Kong (China) and can be administered by the intranasal route [32].

#### 7.1.1. Inactivated Virus

##### CoronaVac

SARS-CoV-2 virus was plaque-purified and cultivated in Vero cells in several passages. The virus was inactivated with b-propiolactone and added to aluminum adjuvant to give the CoronaVac vaccine (Sinovac Biotech, Beijing, China) [32]. This vaccine should be stored at 2 to 8 °C and administered in two doses with an interval of 14 days. CoronaVac shows 50% effectiveness against variant P.1. In the research in Brazil, the efficacy data showed that if the person is infected by the SARS-CoV-2, CoronaVac vaccine offers 100% effectiveness for not getting seriously ill, 78% for preventing mild cases, and 50.38% less risk of becoming ill. The immunizing agent also showed good results against the Delta variant of the coronavirus in the laboratory.

##### CoVaxin

CoVaxin vaccine (BBV152), produced by the Indian laboratory Bharat Biotech, is a whole-virion β-propiolactone-inactivated SARS-CoV-2 vaccine developed with a toll-like receptor 7/8 agonist molecule, and it uses alum as adjuvant. The safety and immunogenicity of BBV152 were tested in several hospitals in India. The trial is registered at ClinicalTrials.gov (NCT04471519). The vaccine (BBV152) must be injected intramuscularly (deltoid muscle) at a volume of 0.5 mL/dose in a two-dose regimen, respecting the interval of two weeks (14 days) between doses. The vaccine had an overall efficacy of 78% in symptomatic cases and 100% in severe cases. In addition, it can be kept at refrigerator temperature, between 2 to 8 °C. The contents of the bottle remain valid for use up to 28 days after opening, provided it is well-stored [36,37].

##### BBIBP-CorV

BBIBP-CorV vaccine, manufactured by Sinopharm and the Wuhan Institute of Virology (WIV)/Beijing Institute of Biological Products, is chemically inactivated whole virus vaccine with propriolactone and produced in Vero cells. It has no viral replication capacity but keeps its proteins intact and generates an immune response with 79% of the original strain. The trial is registered at ClinicalTrials.gov (ChiCTR2000031809; ChiCTR2000032459). Safety and tolerability profile in the two doses administered by the intramuscular route with a 21-day interval between applications was confirmed by study design [34].

### 7.2. Nucleic Acid Vaccines

#### 7.2.1. DNA Vaccine

##### INO-4800

INO-4800 is a plasmid DNA vaccine developed by Inovio Pharmaceuticals (Plymouth Meeting, PA, USA) for preclinical testing within the ferret model of COVID-19. This DNA vaccine encoding the S protein was delivered by electroporation [33]. INO-4800 was able to reduce viral load in non-human primates; rhesus macaque, used as model, that received two doses (1 mg) for 4 weeks of the DNA vaccine showed humoral and cellular immune responses after 13 weeks [38]. The vaccine s+howed effectiveness against both D614 and G614 SARS-CoV-2 variants [39]. The study of safety, tolerability, and immunogenicity of INO-4800 for COVID-19 was tested in healthy volunteers using SARS-CoV-2 spike glycoprotein as antigen.

#### 7.2.2. RNA Vaccine

##### mRNA-1273

A lipid nanoparticle (LNP) encapsulated mRNA vaccine was developed by Moderna (USA); the mRNA encodes the S-2P antigen glycoprotein, which is the S2 subunit with two consecutive substitutions of proline at amino acid positions 986 and 987 [32,33,40]. The mRNA-1273 vaccine induced anti–SARS-CoV-2 immune responses in the first dose, and neutralizing antibodies increased after the second administration of the vaccine in all healthy volunteers in phase 1 clinical trials, irrespective of whether they received a dose of 25 μg, 100 μg, or 250 μg [40]. The vaccine was administered as a 0.5 mL injection by the intramuscular route, using saline solution as a diluent of the doses administered, with 100 μg dose concluded in the phase 3 trial [38]. The adult population (over 18 years) should get a booster dose of an mRNA COVID-19 vaccine (e.g., Moderna or Pfizer–BioNTech) at least 5 months after the last dose of vaccine in the schedule [20].

##### BNT162b1 and BNT162b2 Vaccine

BNT162b1 is a lipid nanoparticle-formulated mRNA vaccine developed by BioNtech SE (Mainz, Germany)/Pfizer (New York, NY, USA) using SARS-CoV-2 proteases (3C-like pro and Mpro) as antigen. It was assessed in three age groups of healthy adults at three different doses (10 µg, 30 µg, or 100 µg) in a randomized study: safety and immunogenicity were demonstrated 14 days after the second administration [40]. After the first administration of the BNT162b2 vaccine in Israel, the efficacy in preventing COVID-19 was 52.4% [41] and 2 to 7 days after the second dose there was 90.5% effectiveness.

The immunogenicity and safety were designed in the Pfizer–BioNTech vaccine on a two-dose schedule in three age groups: ages 5 to 11 years (received two-dose schedule of 10 µg each); ages 2 to 5 years; and ages 6 months to 2 years. Both age groups under 5 years received a lower 3 µg dose for each injection in the phase 2/3 study [42,43]. For booster doses, children over 12 years should receive one at least 5 months after the last dose. Teenagers in age group between 12 to 17 years should get a booster dose of only the Pfizer–BioNTech vaccine. And adults over 18 can receive a booster dose of any mRNA COVID-19 vaccine [20].

### 7.3. Viral Vector Vaccines

Viral vector-based vaccines have been used in gene therapy and include replicating viral vectors and non-replicating viral vectors [32].

#### 7.3.1. Replicating Viral Vectors and Non-Replicating Viral Vectors

##### ChAdOx-1-AZD1222 Vaccine

The AZD1222 vaccine is a chimpanzee viral vector (ChAdOx1) developed at Oxford University and manufactured by AstraZeneca Biopharmaceutical. This vaccine derived from a replication-deficient simian adenovirus vector ChAdOx1 [32], carries sequences encoding the spike protein of SARS-CoV-2 and a tissue plasminogen activator leader sequence [44]. Two doses of AZD1222 should be administered at an interval of 12 weeks between doses, and it must be stored at 2 to 8 °C for no more than six months.

Several viral vectors had been designed to develop vaccines against MERS-CoV using a cloning strategy based on ChAdOx1 with or without sequences encoding the tissue plasminogen activator (tPA) signal peptide [44]. Analyses of humoral and cellular immunogenicity of ChAdOx1 in previous studies on MERS-CoV vaccine candidates demonstrated high responses against the MERS-CoV spike antigen [45]. Induction of virus-neutralizing antibodies was also demonstrated in serum from vaccinated mice [44].

A preliminary study in 2018 of the safety and immunogenicity of the ChAdOx1 MERS vaccine involved administering three different doses in humans by single intramuscular injection. Moderate and severe adverse events were significantly more frequent in the high-dose group (5 × 10^10^ viral particles) than in the intermediate-dose group who received (2.5 × 10^10^ viral particles), but the ChAdOx1 MERS was considered safe and well-tolerated [45]. These single doses elicited both humoral and cellular responses against MERS-CoV [46].

There were mild or moderate local and systemic adverse events similar to those caused by other ChAdOx1-vectored vaccines. Reactogenicity was reported in the first clinical trials of the ChAdOx1 nCoV-19 vaccine, although it was well-tolerated and immunogenic [44].

The same chimpanzee adenovirus (ChAd) vector has also been successfully used for vaccines against other infections such as malaria, HIV, influenza, hepatitis C, tuberculosis, and Ebola. These vaccines show high immunogenicity and safety in humans [45].

A single-dose of the ChAdOx1 MERS vaccine elicits a humoral immune response in rhesus macaques and can neutralize six diverse MERS-CoV strains: (i) EMC/12 (GenBank accession no JX869059); (ii) KSA/15 (GenBank accession no KY688119); (iii) SK/15 (GenBank accession no KU308549); (iv) KSA/18 (GenBank accession no MN723544); (v) C/KSA/13 (GenBank accession no KJ650297); (vi) C/BF/15 (GenBank accession no MG923471). Intranasal vaccination with ChAdOx1 MERS protects BALB/c transgenic mice against the same six strains of MERS-CoV [45,46]. AZD1222 vaccine is effective against the Brazilian variant P1, but not against the South African variant.

##### Human Adenovirus-Ad5 Vaccine or Ad5-nCoV

Ad5-nCoV developed by CanSino Biologics Inc. (Tianjin, China) is an adenovirus type 5 (Ad5) vector with a cytomegalovirus early (CMV) promoter, driving expression of the full-length SARS-CoV-2 spike glycoprotein (S) in the cell [33]. Work with cohorts testing safety have led to Ad5-nCoV not being recommended for immunocompromised individuals [32].

##### Human Adenovirus Type 26 and 5–Sputnik V Vaccine

Sputnik V is based on two adenoviral vectors: AD 26 and AD5. It was developed by the Gamaleya National Center for Epidemiology and Microbiology (Russia). Sequences encoding the spike protein of SARS-CoV-2 were inserted into each vector; the antigenic composition of AD 26 (first dose, 0.5 mL) is different from that of AD 5 (second dose, 0.5 mL) given 21 days later. This strategy circumvents the antigenicity problem. The vaccine should be stored at 2 to 8 °C for no more than six months and can be stored at −20 °C for two years [47].

##### Human Adenovirus (Ad26)–JNJ-78436735 Vaccine

The Ad26 SARS-CoV-2 single-dose vaccine was developed by Johnson and Johnson–J&J (USA), and over a million doses of vaccine from cells were grown in suspension culture in a 1000 liter bioreactor. The non-replicating Ad26 vector (E1/E3 genes deleted) is well-tested for gene therapy and vaccination. It can be transported/stored at 2–8 °C for 6 months or −20 °C for two years [32]. At least 2 months after the first dose it is possible to get a booster dose of J&J/Janssen, and it is also recommended for use with the Pfizer–BioNTech or Moderna (mRNA COVID-19 vaccines) [20].

### 7.4. Protein-Based Vaccines

#### 7.4.1. Protein Subunits and Virus-Like Particles (VLP)

##### NVX-CoV2373 Vaccine

The recombinant SARS-CoV-2 spike protein produced in genetically engineered Sf9 insect cells has been used as a vaccine developed by Novavax (USA). A nanoparticle formulation with saponin-based Matrix-M1 adjuvant was incorporated into the modified protein of this NVX-CoV2373 vaccine [48]. Two doses of 5 µg of NVX-CoV2373 vaccine elicited strong immune responses as assessed by comparing antibody titers with those in convalescent serum from COVID-19 patients [48]. Vaccine administration consists of two doses at a 21-day interval. The ideal storage temperature is 2 to 8 °C for six months, and the vaccine can be stored at −20 °C for two years. It has an efficacy of around 86% against the British variant and 55% against the South African variant.

### 7.5. Other Technological Platforms

#### 7.5.1. LV-SMENP-DC

Dendritic cells (DCs) modified with lentiviral vectors expressing synthetic minigenes encoding domains of viral proteins administered with antigen-specific cytotoxic T lymphocytes (CTLs) have been developed as vaccines by the Shenzhen Geno-Immune Medical Institute [33].

#### 7.5.2. Pathogen-Specific aAPC

Similarly, artificial antigen-presenting cells (aAPCs) modified with lentiviral vectors expressing synthetic minigenes encoding domains of selected viral proteins have also been developed by the Shenzhen Geno-Immune Medical Institute [33].

## 8. COVID-19 Therapy and the Antimicrobial Resistance

Several therapeutic antibodies such as monoclonal antibodies remdesivir, casirivimab, imdevimab, bamlanivimab, and etesevimab are being used to treat COVID-19 infection, inducing moderate or high efficacy against B.1.351, P1, and B.1.1.7 SARS-CoV-2 lineages [20,28]. The monoclonal antibody therapeutics such as sotrovimab, bamlanivimab and etesevimab, and REGEN-COV act mainly on mutations inside the RBD, and they have been used with efficacy against the Omicron variant [20]. Molnupiravir (MK-4482 and EIDD-2801) is a potent ribonucleoside analog that inhibits the replication of SARS-CoV-2 and has been administered by oral route as prevention and treatment of COVID-19 [49]. Paxlovid is the first antiviral drug that is a combination of nirmatrelvir and ritonavir tablets, co-packaged for oral treatment of COVID-19 in adults and pediatric patients (12 years of age) recently approved by the Food and Drug Administration (FDA) [50].

Both vaccines ChAdOx1 nCoV-19 and BNT162b2 use different technological platforms for vaccination booster prime-induced immunity. In particular, BNT162b2 induced high titers of neutralizing antibodies against the B.1.1.7, B.1.351, and P.1 variants [51]. The heterologous prime/booster-inactivated COVID-19 vaccine (CoronaVac) and adenoviral-vectored vaccine (AZD1222) tested for SARSCoV-2 spike receptor-binding domain (RBD) IgG induced higher levels of specific IgG than that of two-dose homologous CoronaVac or AZD1222 vaccines [52]. Aspects related to cellular immunity to SARS-CoV-2, such as virus-specific T cell responses, humoral immunity in the production of neutralizing antibodies, and the natural killer cell transcriptome analysis are key tools for better understanding of immunity in the medium and long term, either by natural infection or by vaccination.

The excessive or inadequate use of antibiotics, antivirals, antifungals, and antiparasitics induces the appearance of microorganisms resistant to the action of drugs, and it can further depress the immune system, leaving it more vulnerable to the emergence of opportunistic infections, as well as to the re-emergence of latent infections such as herpesviruses, for example. The RNA genome of coronaviruses is highly genetically variable, and there is substantial gene recombination leading to frequent emergence of variants. Therefore, the use of antimicrobials in patients with COVID-19 increased the resistance to these drugs, exacerbated by the increased incidence of multi-resistant hospital pathogens and the development of very resistant strains. Consequently, COVID treatment guidelines had been published by the National Institutes of Health (NIH) with the concern, and the recommendation was to mitigate the use inappropriate of antimicrobials because of the incorrect choice of drug, inadequate dosage, and route of administration [20].

## 9. Conclusions

The world is now going through the fourth wave of COVID-19, and it is unclear when or how this viral pandemic will end. Although various technological platforms have been adopted by different countries, the global strategy is to slow down the spread and reduce the incidence of the virus through mass vaccination. The benefits are evident and are reflected in the continuous and sharp drop in cases of COVID-19 and decreases in hospitalizations and numbers of deaths.

Israel is examining cases in which people have had heart inflammation after receiving the Pfizer–BioNTech vaccine. However, no causal relationship between myocarditis and the Pfizer–BioNTech vaccine has been established. New assays have been tested with regard to the interchangeability of the Pfizer–BioNTech vaccine, evaluating the potential benefits for adolescents 12 to 15 years of age and in children 6 months to 11 years old with the expectation of promising results through the results of clinical trials in these age groups [43].

The national health surveillance agency in Brazil recently granted the AstraZeneca vaccine authorization for use. The agreement signed between the private and public partners ensured the technology transfer from the Oxford group that developed the AstraZeneca vaccine to the Immunobiological Technology Institute, Bio-Manguinhos at Oswaldo Cruz Foundation, for national production in Brazil in record time.

The CDC shares SARS-CoV-2 genomic sequence data within accessible databases published by the National Center for Biotechnology Information (NCBI), and the Global Initiative on Sharing Avian Influenza Data (GISAID), which overviews variants/mutations (https://covariants.org/per-variant, accessed on 5 January 2022), has also been used to track variants through a map of tracked variant occurrences and relative variant genome frequencies per region. According to NCBI, up to now, 7236 studies for COVID-19 have been documented in 148 countries, with 614 vaccine studies, 1958 drug studies, and 648 mapped drug names. SARS-CoV-2 genome annotation, sequence record, graphical display, and curated gene records have been shown at Genome Reference Sequence (NC_045512) by NCBI SARS-CoV-2 Resources (https://www.ncbi.nlm.nih.gov/sars-cov-2/, accessed on 5 January 2022). More information about the control of respiratory and related diseases can be found on the website of the National Center for Immunization and Respiratory Diseases (NCIRD) [20].

It will be important to monitor COVID-19 variants through the SARS-CoV-2 surveillance system, particularly as associated with international travel, which favors the spread of the virus and notably emerging variants. Records of new detected variants are being updated all the time. Thus, the tracking of strains of the SARS-CoV-2 virus that are in worldwide circulation can be viewed in real time on the website https://www.gisaid.org/hcov19-variants/ (accessed on 5 January 2022) including a map of tracked variant occurrence with phylodynamic of pandemic coronavirus variant VOC Delta G/478K.V1 (B.1.617.2) first detected in India. Therefore, the importance of genomic surveillance of SARS-CoV-2 is constantly monitored.

Real-time tracking of SARS-CoV-2 evolution and the appearance of new variants involves the use of epidemiological and molecular tools and the website https://nextstrain.org (accessed on 5 January 2022).

Irrespective of the vaccine administered and the technological platform for vaccine production, various characteristics condition the use of a vaccine in the current pandemic: formulation in sterile liquid or in as a lyophilized preparation, whether it is single-use or multidose, dose, stability, route of administration, safety, immunogenicity, and target population [34]. One of the critical points of vaccination is the potential risk for allergic reaction to some types of vaccine components. The current available vaccines do not contain relevant allergens such as egg components, gelatin, latex, or preservatives. All COVID-19 vaccines are metal-free, but both mRNA COVID-19 Pfizer–BioNTech and Moderna vaccines contain polyethylene glycol (PEG).

The active and inactive ingredients of three vaccines (i) Pfizer–BioNTech (mRNA) in different formulations according to each age (5–11 years-orange cap; and over 12 years-gray cap); (ii) Moderna (mRNA); (iii) Janssen (viral vector) have been already described in detail (Appendix C at reference [20]). Patients with contraindications should be referred to an allergist and immunologist for follow-up and should request a consultation from the Clinical Immunization Safety Assessment COVIDvax project [21]. Even so, vaccination is recommended using other technological platforms, such as inactivated (classic), and not new-generation technological platforms, such as mRNA COVID-19 [20]. There are important considerations for COVID-19 vaccination in moderately or severely immunocompromised people, and relevant medical supervision should be established in those population, such as, for example, patients who received organ transplants. Finally, a relevant group that requires relevant care is pregnant and lactating women [20].

Several research centers continue the development of new vaccines against SARS-CoV-2 using advanced technologies focused on new target molecules and new application forms, such as the nasal spray to induce mucosal immunity. Most of these clinical trials are in stages 1, 2, and 3 and plan for scale-up of your mRNA drug, for example.

A novel possibility for the treatment of COVID-19 is the use of monoclonal antibodies. Antibodies such as casirivimab and imdevimab bind to the spike protein and thereby prevent the virus from entering the host cells by recognizing the ACE-2 receptor.

In laboratory tests, SARS-CoV-2 variants containing the L452R or E484K substitution in the spike protein cause a marked reduction in susceptibility to bamlanivimab and possibly also etesevimab and casirivimab [20].

The SARS-CoV-2 spike protein has been central to most of the diagnostic tests and vaccines against COVID-19. The problem is that it can easily mutate, resulting in variants. The nucleocapsid protein is a promising alternative target because it is more strongly conserved among coronaviruses.

Some biopharmaceuticals are evaluating the proposed interchangeability of technologies such as viral vector-based and mRNA-based vaccines [51] or inactivated and adenoviral-vectored vaccines [52] adopted in different laboratories for the development of new viral vaccines due to the emergence of new genomic variants and for the administration of booster doses in the vaccinated population. Thus, a booster dose with a vaccine from a different technological platform (heterologous vaccine) has been shown to have a more effective immune response by increasing the titer of neutralizing antibodies. The course of the pandemic will probably only be controlled by genomic surveillance that tracks variants of SARS-CoV-2. Until that time, Delta continues to be the predominant circulating variant around the world. Finally, the coadministration of COVID-19 vaccines with other vaccines at the same time and the interchangeability of COVID-19 vaccine products—mainly booster doses to improve the immune response—should be evaluated individually in controlled clinical studies.

## Figures and Tables

**Figure 1 ijerph-19-02392-f001:**
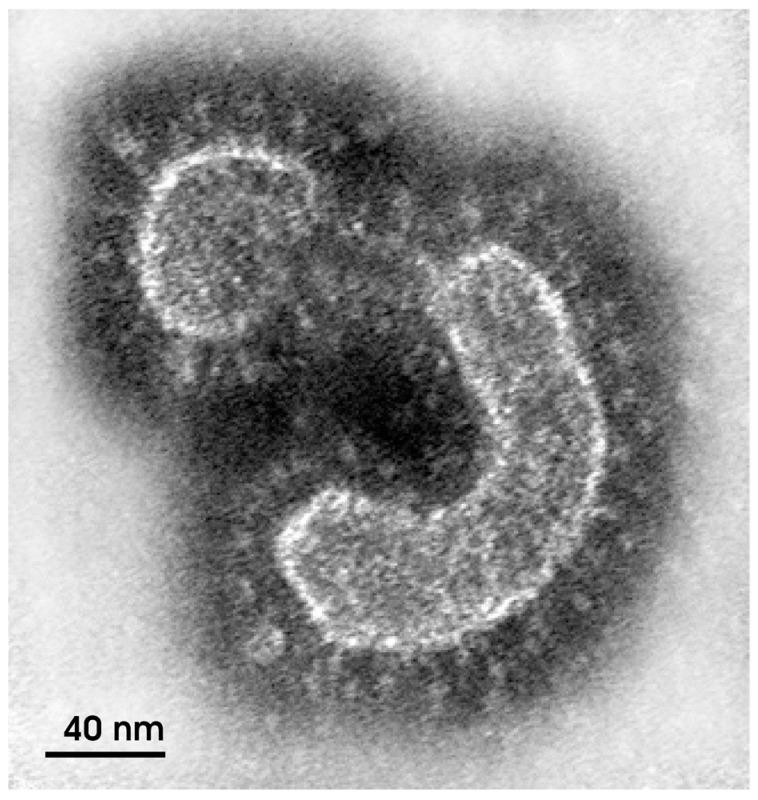
Electron micrograph of a coronavirus particle obtained by negative staining of a clarified suspension of a human fecal sample (Source: Monika Barth/IOC) [3].

**Figure 2 ijerph-19-02392-f002:**
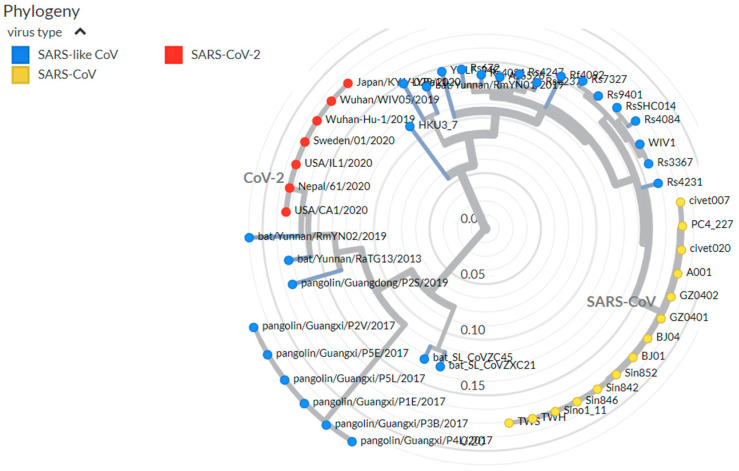
Phylogeny of SARS-like betacoronaviruses including SARS-CoV-2, showing 49 genomes. Adapted from Nextstrain (https://nextstrain.org/groups/blab/sars-like-cov, accessed on 8 February 2022).

**Figure 3 ijerph-19-02392-f003:**
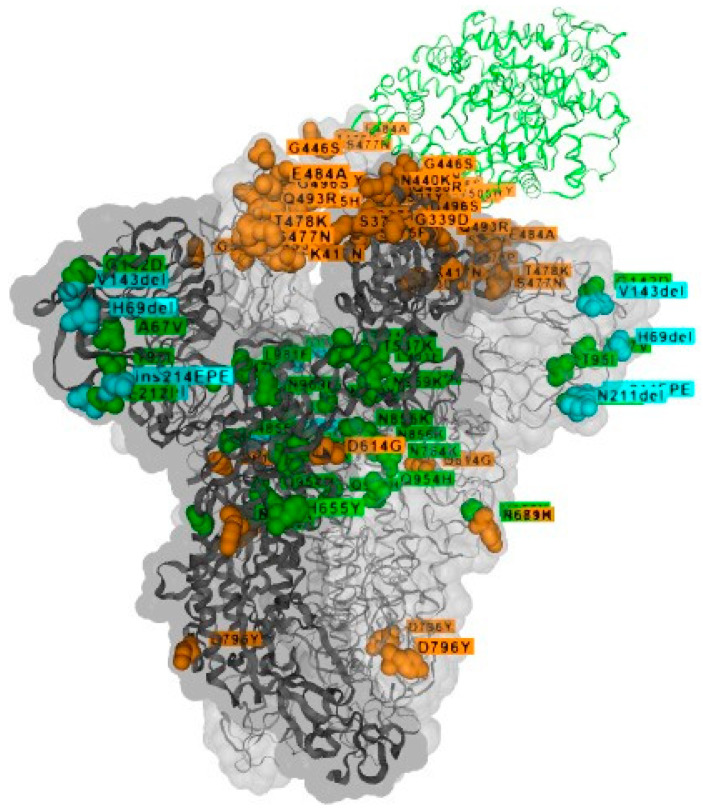
Three-dimensional structure of SARS-CoV-2 spike glycoprotein with positions of amino acid changes in the B.1.1.529+BA lineage (Ὸ). Changes with phenotypic effects are indicated as light orange or cyan for insertions/deletions, while variants without documented phenotypic effects are colored in green. Adapted https://www.gisaid.org/hcov19-variants/ (accessed on 8 February 2022).

**Table 1 ijerph-19-02392-t001:** Characteristics of SARS-CoV-2 Variants of Interest (VOI).

Linage	Named	First Identified	WHO Label	Spike Protein Substitutions
B.1.427	452R	United States (California)	Epsilon (ε)	L452R, D614G
B.1.429	452R	United States (California)	Epsilon (ε)	S13I, W152C, L452R, D614G
B.1.525	484K	United Kingdom/Nigeria	Eta (η)	A67V, 69del, 70del, 144del, E484K, D614G, Q677H, F888L
B.1.526	484K	United States (New York)	Iota (ι)	L5F, (D80G*), T95I, (Y144-*), (F157S*), D253G, (L452R*), (S477N*), E484K, D614G, A701V, (T859N*), (D950H*), (Q957R*)
B.1.617.1	154K	India	Kappa (κ)	(T95I), G142D, E154K, L452R, E484Q, D614G, P681R, Q1071H
B.1.617.3	20A	India	Not named	T19R, G142D, L452R, E484Q, D614G, P681R, D950N
P.2	20J	Brazil (Rio de Janeiro)	Zeta (ζ)	E484K, (F565L*), D614G, V1176F

(*) = detected in some sequences but not all. Adapted from CDC: SARS-CoV-2 Variant Classifications and Definitions (https://www.cdc.gov/coronavirus/2019-ncov/variants/variant-info.html#Interest, accessed on 15 September 2021).

**Table 2 ijerph-19-02392-t002:** Characteristics of SARS-CoV-2 Variants of Concern (VOC).

Linage	Named	First Identified	WHO Label	Spike Protein Substitutions
B.1.1.7	501.Y.V1	United Kingdom	Alpha (α)	69del, 70del, 144del, (E484K*), (S494P*), N501Y, A570D, D614G, P681H, T716I, S982A, D1118H (K1191N*)
B.1.351	501.V2	South Africa	Beta (β)	D80A, D215G, 241del, 242del, 243del, K417N, E484K, N501Y, D614G, A701V
B.1.617.2	478K	India	Delta (δ)	T19R, (G142D*), 156del, 157del, R158G, L452R, T478K, D614G, P681R, D950N
P.1	501Y.V3	Japan/Brazil (Manaus)	Gamma (γ)	L18F, T20N, P26S, D138Y, R190S, K417T, E484K, N501Y, D614G, H655Y, T1027I
B.1.1.529	21K	South Africa	Omicron (Ὸ)	A67V, del69-70, T95I, del142-144, Y145D, del211, L212I, ins214EPE, G339D, S371L, S373P, S375F, K417N, N440K, G446S, S477N, T478K, E484A, Q493R, G496S, Q498R, N501Y, Y505H, T547K, D614G, H655Y, N679K, P681H, N764K, D796Y, N856K, Q954H, N969K, L981F

(*) = detected in some sequences but not all. Adapted from CDC: SARS-CoV-2 Variant Classifications and Definitions (https://www.cdc.gov/coronavirus/2019-ncov/variants/variant-info.html#Interest, accessed on 19 December 2021).

**Table 3 ijerph-19-02392-t003:** Most recent reported occurrences in different countries on the European continent according to tracking of variants (VOC Delta GK (B.1.617.2 + AY)) first detected in India.

Country	Strain Name	Collection Date	GISAID Accession
Southern Europe			
Andorra	hCoV-19/Andorra/AND-235_212521294601_COV-GC/2021	2 September 2021	EPI_ISL_4949311
Italy	hCoV-19/Italy/FVG-UD-451186/2021	18 October 2021	EPI_ISL_5431644
Gibraltar	hCoV-19/Gibraltar/EXP_06092021-1469346-1/2021	5 September 2021	EPI_ISL_4050425
Greece	hCoV-19/Greece/291617/2021	4 October 2021	EPI_ISL_5312764
Monaco	hCoV-19/Monaco/IPP28142/2021	14 September 2021	EPI_ISL_4740992
Portugal	hCoV-19/Portugal/PT18585/2021	6 October 2021	EPI_ISL_5304304
Spain	hCoV-19/Spain/CT-HUVH-EXB20434/2021	19 October 2021	5431911
Turkey	hCoV-19/Turkey/HSGM-E16071/2021	11 October 2021	EPI_ISL_5417865
**Western Europe**			
Austria	hCoV-19/Austria/CeMM16617/2021	30 September 2021	EPI_ISL_4656823
Belgium	hCoV-19/Belgium/AZDelta-2141-11362/2021	20 October 2021	EPI_ISL_5410254
England	hCoV-19/England/SFK-USAFSAM-S6063/2021	18 August 2021	EPI_ISL_4295255
France	hCoV-19/France/ARA-CFD700000136773/2021	11 October 2021	EPI_ISL_5304816
Germany	hCoV-19/Germany/NW-RKI-I-288679/2021	19 October 2021	EPI_ISL_5401385
Ireland	hCoV-19/Ireland/CO-NVRL-M21IRL00168435/2021	13 October 2021	EPI_ISL_5430974
Luxembourg	hCoV-19/Luxembourg/LNS6746091/2021	13 October 2021	EPI_ISL_5394628
Montserrat	hCoV-19/Montserrat/84787/2021	14 August 2021	EPI_ISL_5048422
Reunion	hCoV-19/Reunion/HCL021181330701/2021	4 October 2021	EPI_ISL_5314502
Switzerland	hCoV-19/Switzerland/BE-IFIK-3344-9045/2021	19 October 2021	EPI_ISL_5422230
United Kingdom	hCoV-19/England/MILK-2769536/2021	19 October 2021	EPI_ISL_5407742
**Central-Eastern Europe**			
Bosnia and Herzegovina	hCoV-19/Bosnia and Herzegovina/VFS-UNSA-LMGFI106/2021	30 September 2021	EPI_ISL_5258025
Croatia	hCoV-19/Croatia/10245/2021	21 September 2021	EPI_ISL_5393735
Czech Republic	hCoV-19/Czech Republic/CSQ1588/2021	12 October 2021	EPI_ISL_5316880
Georgia	hCoV-19/Georgia/Tb-SNGS1406/2021	21 September 2021	EPI_ISL_4659774
Kosovo	hCoV-19/Kosovo/CO-00451-XXK000_2946/2021	1 October 2021	EPI_ISL_5429540
Montenegro	hCoV-19/Montenegro/CO-00421_MNE000_2940852102/2021	7 September 2021	EPI_ISL_5104601
Poland	hCoV-19/Poland/WSSEGorzow-21S0916/2021	17 October 2021	EPI_ISL_5408348
Russia	hCoV-19/Russia/ORE-RII-MH36205S/2021	13 October 2021	EPI_ISL_5333181
Serbia	hCoV-19/Serbia/UHB-43087878/2021	5 October 2021	EPI_ISL_5307625
Sint Maarten	hCoV-19/Sint Maarten/SX-RIVM-65014/2021	1 October 2021	EPI_ISL_5429820
Slovakia	hCoV-19/Slovakia/UVZ_PL41_B11_18606/2021	9 October 2021	EPI_ISL_5114073
Slovak Republic	hCoV-19/Czech Republic/CSQ0586/2021	22 August 2021	EPI_ISL_4036828
**Northern Europe**			
Albania	hCoV-19/Albania/un-ChVir25862_21/2021	13 July 2021	EPI_ISL_3125816
Denmark	hCoV-19/Denmark/DCGC-185090/2021	15 October 2021	EPI_ISL_5403927
Estonia	hCoV-19/Estonia/161654/2021	17 September 2021	EPI_ISL_4896194
Finland	hCoV-19/Finland/VI14348/2021	19 September 2021	EPI_ISL_5161299
Iceland	hCoV-19/Iceland/11001/2021	30 August 2021	EPI_ISL_3929959
Latvia	hCoV-19/Latvia/2514/2021	7 July 2021	EPI_ISL_4548626
Lithuania	hCoV-19/Lithuania/LSMULKKGMMK26C175/2021	9 October 2021	EPI_ISL_5304689
Liechtenstein	hCoV-19/Liechtenstein/FL-Risch-21A0804521/2021	7 October 2021	EPI_ISL_5162567
Malta	hCoV-19/Malta/MDxMDH557/2021	10 July 2021	EPI_ISL_3052922
Netherlands	hCoV-19/Belgium/SJ4494691/2021	10 November 2021	EPI_ISL_5428984
Norway	hCoV-19/Norway/Ahus-1938/2021	18 October 2021	EPI_ISL_5431677
Slovenia	hCoV-19/Slovenia/400897/2021	03 October 2021	EPI_ISL_5333468
Sweden	hCoV-19/Sweden/RVFOI21-01498/2021	10 October 2021	EPI_ISL_5323050
**Southeast Europe**			
Bulgaria	hCoV-19/Bulgaria/21BG-EU_006336_Pl58/2021	8 September 2021	EPI_ISL_5417587
North Macedonia	hCoV-19/North Macedonia/43497/2021	31 July 2021	EPI_ISL_3535694
Romania	hCoV-19/Romania/B-480940/2021	11 October 2021	EPI_ISL_5393368
**Eastern Europe**			
Cyprus	hCoV-19/Cyprus/ChVir-LB-210903-8472/2021	28 August 2021	EPI_ISL_3948221
Moldova	hCoV-19/Moldova/un-ChVir25847/2021	6 July 2021	EPI_ISL_3064708
Ukraine	hCoV-19/Ukraine/14-62714/2021	13 July 2021	EPI_ISL_4440044

Adapted from GISAID (https://www.gisaid.org/hcov19-variants/, accessed on 5 January 2022).

**Table 4 ijerph-19-02392-t004:** Most recent reported occurrences in different countries on the Asian continent according to tracking of variants (VOC Delta GK (B.1.617.2 + AY)) first detected in India.

Country	Strain Name	Collection Date	GISAID Accession
Eastern Asia			
China	hCoV-19/Hong Kong/CM21000644/2021	10 October 2021	EPI_ISL_5332916
Japan	hCoV-19/Japan/KMU002014/2021	16 October 2021	EPI_ISL_5419442
Mongolia	hCoV-19/Mongolia/ChVir-LB-211001-4554/2021	23 September 2021	EPI_ISL_4884663
Oman	hCoV-19/Oman/rega-OM-86/2021	5 September 2021	EPI_ISL_4062646
South Korea	hCoV-19/South Korea/KDCA13454/2021	26 September 2021	EPI_ISL_5007927
Taiwan	hCoV-19/Taiwan/TSGH-47/2021	9 September 2021	EPI_ISL4578347
United Arab Emirates	hCoV-19/United Arab Emirates/AZ-USAFSAM-S4144/2021	23 June 2021	EPI_ISL_3048168
**Southeast Asia**			
Cambodia	hCoV-19/Cambodia/826388/2021	10 October 2021	EPI_ISL_5260555
Indonesia	hCoV-19/Indonesia/JB-GS-WJHL-ITB-W102/2021	6 October 2021	EPI_ISL_532847
Iran	hCoV-19/Iran/Ardakan-NIC-S5/2021	3 September 2021	EPI_ISL_4893538
Jordan	hCoV-19/Jordan/RMSR8S4/2021	15 October 2021	
Malaysia	hCoV-19/Malaysia/C19UMB396/2021	14 October 2021	EPI_ISL_5428869
Singapore	hCoV-19/Singapore/6972/2021	17 October 2021	EPI_ISL_5421871
Thailand	hCoV-19/Thailand/Narathiwat_SEQ18255/2021	7 October 2021	EPI_ISL_5416195
**South Asia**			
Bangladesh	hCoV-19/Bangladesh/icddrb-CU-CGH-4883/2021	2 October 2021	EPI_ISL_5161318
Bahrain	hCoV-19/Bahrain/21921634218/2021	2 October 2021	EPI_ISL_5032888
India	hCoV-19/India/GA-CDFD-C62748/2021	8 October 2021	EPI_ISL_5003985
Maldives	hCoV-19/Maldives/MAV84800/2021	2 October 2021	EPI_ISL_5061914
Myanmar	hCoV-19/Myanmar/DSMRC022/2021	26 August 2021	EPI_ISL_5424999
Nepal	hCoV-19/Nepal/NPHL-S-232/2021	22 September 2021	EPI_ISL_5064726
Pakistan	hCoV-19/Pakistan/NIH-B21-S19/2021	10 October 2021	EPI_ISL_5427755
Philippines	hCoV-19/Philippines/PH-PGC-59084/2021	14 July 2021	4741922
Sri Lanka	hCoV-19/Sri Lanka/idea_uoc_00099/2021	5 October 2021	EPI_ISL_5314134
Vietnam	hCoV-19/Vietnam/NHTD-OUCRU1235/2021	24 September 2021	EPI_ISL_4969171
**Central Asia**			
Afghanistan	hCoV-19/Afghanistan/2870x0209_23909/2021	24 May 2021	EPI_ISL_4572808
Iraq	hCoV-19/Iraq/DSeq-C38/2021	13 June 2021	EPI_ISL_4574554
Kazakhstan	hCoV-19/Kazakhstan/1742/2021	19 August 2021	EPI_ISL_4470505
Uzbekistan	hCoV-19/Uzbekistan/Tashkent-CGB-38/2021	23 July 2021	EPI_ISL_3668632
**Western Asia**			
Azerbaijan	hCoV-19/Azerbaijan/Aghayev-07-02/2021	14 July 2021	EPI_ISL_2932742
Armenia	hCoV-19/Armenia/IMB8-6/2021	5 August 2021	EPI_ISL_3543642
Israel	hCoV-19/Israel/NRL_12165/2021	26 August 2021	EPI_ISL_4505006
Jordan	hCoV-19/Jordan/RMSR8S4/2021	15 October 2021	EPI_ISL_5323884
Lebanon	hCoV-19/Lebanon/LAU-CVD-112/2021	16 July 2021	EPI_ISL_3233231
Qatar	hCoV-19/Qatar/DA-USAFSAM-S6014/2021	16 August 2021	EPI_ISL_42955239
Timor-Leste	hCoV-19/Timor-Leste/TL1028/2021	8 August 2021	EPI_ISL_4761841
Kuwait	hCoV-19/Kuwait/Jaber2120390135/2021	26 September 2021	EPI_ISL_4891369

Adapted from GISAID (https://www.gisaid.org/hcov19-variants/, accessed on 5 January 2022).

**Table 5 ijerph-19-02392-t005:** Most recent reported occurrences in different countries on the American continent according to tracking of variants (VOC Delta GK (B.1.617.2 + AY)) first detected in India.

Country	Strain Name	Collection Date	GISAID Accession
North America			
British Virgin Islands	hCoV-19/British Virgin Islands/81148/2021	27 July 2021	EPI_ISL_3655569
Canada	hCoV-19/Canada/ON-PHL-21-34650/2021	1 October 2021	EPI_ISL_5077231
Cayman Islands	hCoV-19/Cayman Islands/43670/2021	14 October 2021	EPI_ISL_5239459
Martinique	hCoV-19/Martinique/IPP29690/2021	5 October 2021	EPI_ISL_5409955
Mexico	hCoV-19/Mexico/CMX-INMEGEN-32-227/2021	13 October 2021	EPI_ISL_5410229
Turks and Caicos Islands	hCoV-19/Turks and Caicos Islands/78223/2021	12 July 2021	EPI_ISL_3642830
USA	hCoV-19/USA/CT-YPL-21-117778/2021	19 October 2021	EPI_ISL_5447321
U.S. Virgin Islands	hCoV-19/U.S. Virgin Islands/USVI-Yale-10749-/2021	13 September 2021	EPI_ISL_5196074
**South America**			
Argentina	hCoV-19/Argentina/PAIS-E0431/2021	12 October 2021	EPI_ISL_5332852
Brazil	hCoV-19/Brazil/SP-HIAE-ID892/2021	6 October 2021	EPI_ISL_4926944
Bonaire	hCoV-19/Bonaire/BQ-RIVM-63933/2021	5 October 2021	EPI_ISL_5429667
Chile	hCoV-19/Chile/LR-UACH-00326/2021	16 October 2021	EPI_ISL_5421080
Colombia	hCoV-19/Colombia/ANT-LDSP918/2021	23 September 2021	EPI_ISL_5143467
Curacao	hCoV-19/Curacao/CW-RIVM-63860/2021	3 October 2021	EPI_ISL_5209099
Ecuador	hCoV-19/Ecuador/USFQ-2139/2021	23 September 2021	EPI_ISL_5073650
French Guiana	hCoV-19/French Guiana/IPP29099/2021	14 September 2021	EPI_ISL_5054141
Guatemala	hCoV-19/Guatemala/INC-LNS-048/2021	14 September 2021	EPI_ISL_5169140
Paraguay	hCoV-19/Paraguay/233072/2021	29 August 2021	EPI_ISL_4259639
Peru	hCoV-19/Peru/LIM-UPCH-1309/2021	25 September 2021	EPI_ISL_5147486
Venezuela	hCoV-19/Venezuela/Mir8310/2021	7 July 2021	EPI_ISL_3298770
**Central America**			
Antigua and Barbuda	hCoV-19/Antigua and Barbuda/84387/2021	18 August 2021	EPI_ISL_5048415
Anguilla	hCoV-19/Anguilla/83840/2021	13 August 2021	EPI_ISL_5048447
Aruba	hCoV-19/Aruba/AW-RIVM-63965/2021	3 October 2021	EPI_ISL_5429689
Bahamas	hCoV-19/Bahamas/48034/2021	8 August 2021	EPI_ISL_4237137
Barbados	hCoV-19/Barbados/79988/2021	25 July 2021	EPI_ISL_3655568
Belize	hCoV-19/Belize/CML-109/2021	30 June 2021	EPI_ISL_4296413
Brunei	hCoV-19/Brunei/29/2021	17 August 2021	EPI_ISL_3681508
Costa Rica	hCoV-19/Costa Rica/INC-1343-705546/2021	1 October 2021	EPI_ISL_5262753
Dominican Republic	hCoV-19/Dominican Republic/49666/2021	14 August 2021	EPI_ISL_4220384
El Salvador	hCoV-19/El Salvador/INC-LNSP-043/2021	2 August 2021	EPI_ISL_5158232
Grenada	hCoV-19/Grenada/81073/2021	26 July 2021	EPI_ISL_3688254
Guadeloupe	19/Guadeloupe/IPP29637/2021	28 September 2021	EPI_ISL_5409931
Haiti	hCoV-19/Haiti/63270211/2021	27 July 2021	EPI_ISL_4028961
Honduras	hCoV-19/Honduras/1292739/2021	31 July 2021	EPI_ISL_4029392
Jamaica	hCoV-19/Jamaica/82372/2021	23 July 2021	EPI_ISL_4055937
Panama	hCoV-19/Panama/GMI-PACOVIDVIRO806/2021	29 April 2021	EPI_ISL_5196333
Puerto Rico	hCoV-19/Puerto Rico/PR-Yale-11110-/2021	21 September 2021	EPI_ISL_5195974
Saint Barthelemy	hCoV-19/Saint Barthelemy/IPP28734/2021	21 September 2021	EPI_ISL_5053995
Saint Lucia	hCoV-19/Saint Lucia/80752/2021	26 July 2021	EPI_ISL_3655575
Saint Vincent and the Grenadines	hCoV-19/Saint Vincent and the Grenadines/82973/2021	8 August 2021	EPI_ISL_4055941
Suriname	hCoV-19/Suriname/SR-680/2021	17 September 2021	EPI_ISL_5018013
Trinidad and Tobago	hCoV-19/Trinidad and Tobago/84746/2021	13 August 2021	EPI_ISL_5048491

Adapted from GISAID (https://www.gisaid.org/hcov19-variants/, accessed on 5 January 2022).

**Table 6 ijerph-19-02392-t006:** Most recent reported occurrences in different countries on the Oceania continent according to tracking of variants (VOC Delta GK (B.1.617.2 + AY)) first detected in India.

Country	Strain Name	Collection Date	GISAID Accession
Australia	hCoV-19/Australia/QLD2471/2021	20 October 2021	EPI_ISL_5416529
Fiji	hCoV-19/Fiji/FJ620/2021	9 July 2021	EPI_ISL_4820112
Guam	hCoV-19/Guam/Yigo-USAFSAM-S4904/2021	26 July 2021	EPI_ISL_4078922
New Zealand	hCoV-19/New Zealand/21MV2279/2021	16 October 2021	EPI_ISL_5348864

**Table 7 ijerph-19-02392-t007:** Most recent reported occurrences in different countries on the African continent according to tracking of variants (VOC Delta GK (B.1.617.2 + AY) first detected in India.

Country	Strain Name	Collection Date	GISAID Accession
East Africa			
Burundi	hCoV-19/Burundi/UG893/2021	28 July 2021	EPI_ISL_4949209
Ethiopia	hCoV-19/Ethiopia/AFCPH_IDSWH_29222/2021	16 August 2021	EPI_ISL_4063320
Kenya	hCoV-19/Kenya/SS2359/2021	16 September 2021	EPI_ISL_4739516
Malawi	hCoV-19/Malawi/MLW-00072/2021	5 August 2021	EPI_ISL_3770691
Mozambique	hCoV-19/Mozambique/CERI-KRISP-K025862/2021	7 August 2021	EPI_ISL_5425734
Rwanda	hCoV-19/Rwanda/NRL/DB17645-09/2021	2 September 2021	EPI_ISL_4632886
Seychelles	hCoV-19/Seychelles/SS2729/2021	31 August 2021	EPI_ISL_4880794
Uganda	hCoV-19/Uganda/CERI-UVRI-K022877/2021	12 August 2021	EPI_ISL_4548467
Zambia	hCoV-19/Zambia/ZMB-119161/2021	2 September 2021	EPI_ISL_4513785
Zimbabwe	hCoV-19/Zimbabwe/CERI-KRISP-K021559/2021	26 July 2021	EPI_ISL_3730448
**Western Africa**			
Benin	hCoV-19/Benin/481519/2021	23 July 2021	EPI_ISL_4566987
Ghana	hCoV-19/Ghana/TRA-2013/2021	14 September 2021	EPI_ISL_4919699
Guinea	hCoV-19/Guinea/CERFIG-25618/2021	8 July 2021	EPI_ISL_4205871
Liberia	hCoV-19/Liberia/LIB-0094/2021	10 July 2021	EPI_ISL_3547691
Mali	hCoV-19/Mali/CICM1649/2021	11 August 2021	EPI_ISL_5429401
Nigeria	hCoV-19/Nigeria/NCDC-NR563/2021	22 September 2021	EPI_ISL_4743214
Sierra Leone	hCoV-19/Sierra Leone/CPHRL-22/2021	8 September 2021	EPI_ISL_4635451
Senegal	hCoV-19/Senegal/SN-IRVAC-172/2021	1 September 2021	EPI_ISL_4880794
Togo	hCoV-19/Togo/NMIMR-21-TGS-457/2021	31 July 2021	EPI_ISL_3915134
**North Africa**			
Algeria	hCoV-19/Algeria/76465/2021	1 September 2021	EPI_ISL_5052212
Egypt	hCoV-19/Egypt/ARMY-EVA-Pharma-Wave4-012/2021	4 September 2021	EPI_ISL_4750218
Morocco	hCoV-19/Morocco/20358089/2021	3 September 2021	EPI_ISL_4741167
South Sudan	hCoV-19/South Sudan/UG702/2021	24 July 2021	EPI_ISL_3546339
Tunisia	hCoV-19/Tunisia/34737/2021	21 May 2021	EPI_ISL_2907578
**Central Africa**			
Angola	hCoV-19/Angola/CERI-KRISP-K024658/2021	12 August 2021	EPI_ISL_4474366
Democratic Republic of the Congo	hCoV-19/DRC/INRB-RDC-548/2021	12 July 2021	EPI_ISL_3086923
Equatorial Guinea	hCoV-19/Equatorial Guinea/77627/2021	30 August 2021	EPI_ISL_4601605
Gabon	hCoV-19/Gabon/CERMEL-JJ0692/2021	30 September 2021	EPI_ISL_5031211
Papua New Guinea	hCoV-19/Papua New Guinea/PNG3575/2021	12 September 2021	EPI_ISL_4948709
Republic of the Congo	hCoV-19/Congo/FCRM-81-A1/2021	20 September 2021	EPI_ISL_4724412
**Southern Africa**			
Botswana	hCoV-19/Botswana/R33B87_BHP_AAC6293/2021	27 September 2021	EPI_ISL_5248613
Burkina Faso	hCoV-19/Burkina Faso/CV1935/2021	21 April 2021	EPI_ISL_4255149
Central African Republic	hCoV-19/Central African Republic/ JXCDC-85/2021	21 January 2022	EPI_ISL_3398832
Eswatini	hCoV-19/Eswatini/N13458/2021	26 July 2021	EPI_ISL_4301841
Gambia	hCoV-19/Gambia/49615/2021	12 July 2021	EPI_ISL_3132315
Mauritius	hCoV-19/Mauritius/358802/2021	23 September 2021	EPI_ISL_5260440
Mayotte	hCoV-19/Mayotte/HCL021178419401/2021	28 September 2021	EPI_ISL_5313309
Namibia	hCoV-19/ Namibia/N32192/2021	9 February 2022	EPI_ISL_4253775
South Africa	hCoV-19/South Africa/Tygerberg_2860/2021	8 October 2021	EPI_ISL_5264713

Adapted from GISAID (https://www.gisaid.org/hcov19-variants/, accessed on 5 January 2022).

**Table 8 ijerph-19-02392-t008:** Most recent reported occurrences in different countries according to tracking of variants (VUM GH/490R (B.1.640 + B.1.640*)) first detected in Congo/France.

Country	Strain Name	Collection Date	GISAID Accession
Southern Europe			
France	hCoV-19/France/ARA-HCL021230857901/2021	5 January 2022	EPI_ISL_8376567
United Kingdom	hCoV-19/England/PHEC-3X054XB1/2021	5 January 2022	EPI_ISL_8380761
Luxembourg	hCoV-19/Luxembourg/LNS2635248/2021	4 January 2022	EPI_ISL_8346694
Germany	hCoV-19/Germany/HH-hpi-p11204/2021	4 January 2022	EPI_ISL_8338416
Canada	hCoV-19/Canada/ON-PHL-21-48197/2021	4 January 2022	EPI_ISL_8324385
Belgium	hCoV-19/Belgium/MBLG-CTMAOT12212008/2021	3 January 2022	EPI_ISL_8317146
Switzerland	hCoV-19/Switzerland/SG-CLM-12174874/2021	24 December 2021	EPI_ISL_8038963
Kenya	hCoV-19/Kenya/SS4231/2021	20 December 2021	EPI_ISL_7868871
Italy	hCoV-19/Italy/LAZ-INMI-2887/2021	17 December 2021	EPI_ISL_7780056
Reunion	hCoV-19/Reunion/HCL021215597001/2021	13 December 2021	EPI_ISL_7611111
Indonesia	hCoV-19/Indonesia/JK-NIHRD-WGS13396/2021	11 December 2021	EPI_ISL_7547548
USA	hCoV-19/USA/IL-S21WGS5813/2021	10 December 2021	EPI_ISL_7493953
India	hCoV-19/India/DL-ILBS-WGS584/2021	8 December 2021	EPI_ISL_7379386
Republic of the Congo	hCoV-19/Congo/RC-194/2021	2 December 2021	EPI_ISL_6951080
Ghana	hCoV-19/Ghana/TRA-2159/2021	1 December 2021	EPI_ISL_6944033
Spain	hCoV-19/Spain/CT-HUVH-E44748/2021	15 November 2021	EPI_ISL_6268734
Netherlands	hCoV-19/Netherlands/GR-RIVM-68090/2021	12 November 2021	EPI_ISL_6230267
South Africa	hCoV-19/South Africa/NICD-R11147/2021	11 August 2021	EPI_ISL_3342437
Russia	hCoV-19/Russia/SPE-RII-MH17346S/2021	3 June 2021	EPI_ISL_2385305

Adapted from GISAID (https://www.gisaid.org/hcov19-variants/, accessed on 5 January 2022).

**Table 9 ijerph-19-02392-t009:** Technological platforms for SARS-CoV-2 vaccine candidates licensed or under development.

Technological Platform	Vaccine Name	Biopharmaceutical/Institution	Target	Efficacy *	Dose/Interval	Stored
Virus inactivated	CoronaVac	SinoVac Biotech; Instituto Butantan	Whole virus	50–90%	2/14 days	2−8 °C
	BBIBP-CorV	Sinopharm	Whole virus	79%	2/21 days	2−8 °C
	CoVaxin (BBV152 vaccine)	Bharat Biotech	Whole virus	78–100%	2/14 days	2−8 °C
DNA vaccine	INO-4800	Inovio Pharmaceuticals	Spike	95%	2/4 weeks	room temperature stock
RNA vaccine	mRNA-1273	Moderna	Spike	95%	2/28 days	−20 °C (6 months)
	BNT162	BioNTech/Pfizer	3CLpro, NSP5, Mpro **	90%	2/21 days	2−8 °C (5 days) or −70 °C (6 months)
Viral vector (non-replicating)	AZD1222 (ChAdOx-1 nCoV-19)	University of Oxford/Astra Zeneca; Oswaldo Cruz Foundation-Bio-Manguinhos	Spike	82%	2/12 weeks	2−8 °C
	Ad5-nCoV	CanSino Biologics	Spike	90%	1 dose	2–8 °C
	Ad26 (JNJ-78436735 vaccine)	Johnson & Johnson–Janssen	Undisclosed	72%	1 dose	2–8 °C (6 months) or −20 ° C (two years)
Viral vector (replicating)	Sputnik V vaccine	Gamaleya Research Institute of Epidemiology and Microbiology	Spike	91%	2/21 days	2–8 °C (6 months) or −20 °C (two years)
Virus-like particles (VLP)	NVX-CoV2373	Novavax	Spike (prefusion)	86%	2/21 days	2–8 °C (6 months) or −20 °C (two years)

* Vaccine efficacy data can be different, as the protocols are not always the same, as well as the behavior of the population, which changes from country to country and even within the same country, depending on the situation of the pandemic. ** SARS-CoV-2 3C-like protease (3CLpro), NSP5, SARS-CoV-2 main protease (Mpro). Sources [32,35].

## Data Availability

The authors used as their database various sources of literature and software—https://www.cdc.gov/coronavirus/2019-ncov/variants/variant-info.html#Interest (accessed on 5 January 2022), https://www.gisaid.org/hcov19-variants/ (accessed on 5 January 2022), and https://nextstrain.org (accessed on 5 January 2022)—to update the data in real time.

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
