# Peer review of "Classical and Next-Generation Vaccine Platforms to SARS-CoV-2: Biotechnological Strategies and Genomic Variants"

_ijerph, 2022, doi:10.3390/ijerph19042392_

Round 1

Reviewer 1 Report

In this review article Rachel Siqueira et al. described in great detail the characteristics and physiology of SARS-Cov-2. Unfortunately the most interesting section of the manuscript (7. Classical and Next-generation vaccine platforms), which relates directly to the title of this work, is shallow and contributes little to the field.

Many new areas are left behind, such as the use of recombinant RBD domain as antigen or the incorporation of small peptides and or fragments from the Spike glycoprotein in nanoparticles.

This review will benefit greatly by discussing in greater detail the strategies mentioned in the section 7, and including new ones currently under development.

Author Response

Information and new data on vaccine strategies have been enriched and updated throughout the text.

In addition: 

Figure 1: We have modified the figure to include Bar (nm)

Figure 2: Phylogeny of SARS-like betacoronaviruses including SARS-CoV-2, showing 49 genomes. Adapted https://nextstrain.org/groups/blab/sars-like-cov .

Figure 3: Three-dimensional structure of SARS-CoV-2 spike glycoprotein with positions of amino acid changes in the B.1.1.529+BA lineage (Ὸ).

Reviewer 2 Report

This is a factual account of `Classical and next-generation vaccine platforms to SARS-CoV-2: biotechnological strategies and genomic variants` The data are useful for the readership, and is well written. Personally, a more critical review would have added value to the ms, If possible please do so.

Author Response

More critical information was inserted regarding the administration of the COVID-19 vaccine in association with the use of other vaccines in the immunization schedule, interchangeability of technological platforms mainly in the booster dose to increase immunity (heterologous vaccine), considerations of vaccination in specific groups as allergic individuals, transplanted, immunosuppressed, pregnant and lactating.

In addition:

Figure 1: We have modified the figure to include Bar (nm)

Figure 2: Phylogeny of SARS-like betacoronaviruses including SARS-CoV-2, showing 49 genomes. Adapted https://nextstrain.org/groups/blab/sars-like-cov .

Figure 3: Three-dimensional structure of SARS-CoV-2 spike glycoprotein with positions of amino acid changes in the B.1.1.529+BA lineage (Ὸ).

Reviewer 3 Report

The review entitled “Classical and Next-Generation Vaccine Platforms to SARS-CoV-2: Biotechnological Strategies and Genomic variants” discusses vaccine platforms against SARS Corona Viruses. However, the manuscript lacks the necessary systematic flow of information and the through literature review on vaccine development which render it unacceptance in its current form.  

Major

  1. Although the title gives the idea that the review will be focusing on vaccine platforms, the vaccines platforms are mentioned briefly in point 7 with inefficient details and not reviewing the platforms well enough. For example, in section 7.4 (protein-based vaccines), has 2 subcategories (7.41 and 7.4.1.1) and only mention one vaccine “NVX-CoV2373” despite many in clinical trials.

  1. The Language and the flow of information in the article are poor and irritating to the reader and there is only one figure in the entire manuscript (an EM figure of the virus), other figures facilitating illustration of the review’s idea should be added.

  1. The references need to be revised, for example reference 34; is a 2-page article and mentioned about 14 times in different vaccine platforms which makes it difficult for the reader to track the information presented in this review article.

Minor

  1. Line 37 and 38; please use right punctuation (add commas instead of multiple ands)

  1. Line 40; please add appropriate reference for Andersen et al., 2020

  1. Line 50; please don’t keep re-defining abbreviations (severe acute respiratory syndrome coronavirus 2 - SARS-CoV-2,). Define them once and then use the abbreviation in the rest of the manuscript

  1. Paragraph (Lines 67-72); these are enumerated in the abstract, please mention the groups only in the abstract and enumerate examples in the introduction paragraph.

Author Response

  1. Although the title gives the idea that the review will be focusing on vaccine platforms, the vaccines platforms are mentioned briefly in point 7 with inefficient details and not reviewing the platforms well enough. For example, in section 7.4 (protein-based vaccines), has 2 subcategories (7.41 and 7.4.1.1) and only mention one vaccine “NVX-CoV2373” despite many in clinical trials.

The topic 7.4.1. Protein subunits and virus-like particles (VLP) correspond to (item 7.4) the Protein-Based Vaccines as technology platform which has as an example the NVX-CoV2373 vaccine enumerated as 7.4.1.1.

  1. The Language and the flow of information in the article are poor and irritating to the reader and there is only one figure in the entire manuscript (an EM figure of the virus), other figures facilitating illustration of the review’s idea should be added.

We have improved the language and include new relevant information, in this way, to try to make the reading more fluid and interesting to the reader.

  1. The references need to be revised, for example reference 34; is a 2-page article and mentioned about 14 times in different vaccine platforms which makes it difficult for the reader to track the information presented in this review article.

Following the reviewer’s suggestion, some references were excluded and updated. Regrading the mentioned reference was published in Nature Reviews | Drug Discovery (2020), and reports the diversity of technological platforms, the central object of this review justifying the number of citations throughout the text.

Minor 

  1. Line 37 and 38; please use right punctuation (add commas instead of multiple ands)

Done

  1. Line 40; please add appropriate reference for Andersen et al., 2020

Done. The reference Anderson et al, 2020 become reference 1, so the numerical order of bibliographic citations has been changed up to 7.

  1. Line 50; please don’t keep re-defining abbreviations (severe acute respiratory syndrome coronavirus 2 - SARS-CoV-2,). Define them once and then use the abbreviation in the rest of the manuscript

Done

  1. Paragraph (Lines 67-72); these are enumerated in the abstract, please mention the groups only in the abstract and enumerate examples in the introduction paragraph.

Examples of each technology platform have been now included according to the abstract.

Round 2

Reviewer 1 Report

Authors have addressed most of the comments.

Reviewer 3 Report

the authors addressed my comments